# A global scoping review of the circumstances of care seeking for abortion later in pregnancy

**Laura E. Jacobson**[1], **Blair G. Darney**[2], **Heidi Bart Johnston**[1], **Bela Ganatra**[1] *

**1** UNDP-UNFPA-UNICEF-WHO-World Bank Special Programme of Research, Development and Research Training in Human Reproduction (HRP), Department of Sexual and Reproductive Health and Research, World Health Organization, Geneva, Switzerland, **2** Dept Obstetrics & Gynecology, Oregon Health & Science University, Portland, Oregon, United States of America

* ganatrab@who.int

**Data Availability Statement:** Data are publicly available, published manuscripts.

## Abstract

Understanding the circumstances of abortions later in pregnancy provides insight about the barriers and delays to timely care. Limited synthesized information is available on these circumstances, especially from low and middle incomes countries. Reviewing what is reported in the literature about the circumstances of abortion later in pregnancy and the methodological approaches used to study this is needed to reveal evidence gaps. The purpose of this study is to describe what is documented and methodological approaches used in existing literature on the circumstances and characteristics associated with seeking care for abortion later in pregnancy reported in population and facility-based studies. We conducted a scoping review of peer-reviewed research in OVID/PubMed, Embase, Scopus, SocIndex, and LILACs from 2007–2024 that described the circumstance, sociodemographic characteristics, population- or facility-based proportion of abortion later in pregnancy (≥12 weeks of gestation or "second trimester") reported in the literature. We screened 2598 records by title and/or abstract and 668 of those by full text. We included 78 studies that described the circumstances around seeking care for abortion later in pregnancy from qualitative data (12 studies); included information on associated characteristics from quantitative data (15 studies); reported a population- (17 studies) or facility-based (45 studies) proportion of abortion later in pregnancy. Prominent themes included health system challenges, late pregnancy recognition, financial challenges, and delayed decision making. Low economic status and adolescence were commonly associated characteristics. Population and facility-based studies lacked standardization when reporting durations of gestation. Facility studies reported a wide variety of populations and number of facilities. Circumstances surrounding abortions later in pregnancy include health system challenges, late pregnancy recognition, financial issues, and delayed decision-making, which intersect to compound and extend delays. More research guided by clear methods and standard definitions when reporting on population and facility-based proportions of abortions later in pregnancy is needed to reveal evidence gaps and better inform policies and programs.

**Funding:** This work was supported by the UNDP-UNFPA-UNICEF-WHO-World Bank Special Programme of Research, Development and Research Training in Human Reproduction (HRP), a cosponsored programme executed by the World Health Organization (WHO) (to author BG). The funders had no role in study design, data collection and analysis, decision to publish, or preparation of the manuscript. The views expressed in this article are those of the authors and do not necessarily represent the views of, and should not be attributed to, the World Health Organization.

**Competing interests:** LEJ has nothing to disclose. BGD is a member of the Board of Directors of the Society of Family Planning (SFP) and CISIDAT (Health Research Consortium, Mexico). She is a Deputy Editor at Contraception and a committee member at the American College of Obstetrics and Gynecology (ACOG), activities for which she receives honoraria. She is also supported by a Garcia-Robles COMEXUS-Fulbright award, Mexico, 2023-2024. HBJ has nothing to disclose. BG has nothing to disclose.

## Background

Abortion is an essential sexual and reproductive health care service. Globally, induced abortion is common: an estimated 73.3 million abortions occur annually worldwide both in settings where abortion law is liberal and where it is restrictive [1]. Legal restrictions to abortion reduce access to safe abortion care [2]. Abortion can be safely provided throughout pregnancy [3], although the provision of both medical and surgical abortion is simpler in early pregnancy. In high income countries, over 90% of all abortions occur before 13 weeks of gestation, and more than two-thirds of abortions occur before nine weeks of gestation [4]. Abortion is necessary later in pregnancy due to delayed pregnancy recognition, delays in access to care due stigma and logistics, and receiving new medical information [5, 6]. Known barriers to accessing abortion care later in pregnancy include limited availability of trained providers and facilities authorized to provide services, high costs, abortion-related stigma, and legal restrictions [7–9]. However, there is limited synthesized information available on the circumstances of those seeking abortion later in pregnancy, especially from low and middle incomes countries.

The purpose of this scoping review was to describe the circumstances associated with seeking care for abortion later in pregnancy and how abortions later in pregnancy are reported in the literature. These questions are important to provide insights into the complex factors correlated with seeking care for abortion later in pregnancy and to identify gaps in the current state of reporting and documentation of abortion. This knowledge can inform healthcare professionals, policymakers, and researchers as they develop evidence-based interventions to improve the accessibility and quality of reproductive healthcare services. To address the knowledge gap, we conducted a scoping review to identify what is reported in the literature about the circumstances of abortion later in pregnancy (≥12 weeks of gestation or labeled as "second trimester"), the methodological approaches used to study and report this, and key evidence gaps to be addressed in future research.

## Methods

We conducted a scoping review informed by Arksey & O'Malley [10]. We described the circumstances of an abortion later in pregnancy, meaning the contextual factors involved in or reasons for needing abortion later in pregnancy as well as individual sociodemographic or other characteristics. We anticipated finding varied types of evidence from diverse study designs and our interest was in documenting this variability. We also summarized available reports of the proportion of abortions later in pregnancy (≥12 weeks of gestation or labeled as "second trimester") from both population- and facility-based studies. We conducted this review following PRISMA-ScR guidelines [11].

Author LJ collaborated with an academic public health librarian to design and run the database search using five bibliographic databases: Scopus, Ovid/PubMed, Embase, SocIndex, and LILACs to identify articles published between January 2007-April 2024. The scoping review considered publications reporting quantitative and qualitative analyses on incidence of experience of induced abortion, denial of abortion, and/or post-abortion care for complication after induced abortion later in pregnancy from any country that were published in peer-reviewed and grey literature. Each search was customized for the database and sought to capture two main concepts: 1. circumstances surrounding seeking care for abortion later in pregnancy and 2. proportion of abortions reported from population or facility-data occurring later in pregnancy. The major search concepts are displayed in **Table 1** and the bibliographic database search strategy for OVID/PubMed can be found in **S1 Table,** all other search strategies contained the same terms.

**Table 1. Major concepts utilized in search, in various combinations, utilizing subject terms and syntax as appropriate to database.**

| Concept | Syntax |
|---|---|
| Abortion | abortion* OR pregnancy termination OR termination of pregnancy OR miscarriage* OR postabortion OR post-abortion |
| Reproductive health | menstruation OR menstrual OR pregnancy OR reproductive rights OR reproductive health |
| Procedural abortion | (dilation and curettage) OR (d and c) OR dilation OR (surgical AND (evacuation OR termination)) OR (curettage OR (vacuum AND curettage) OR (vacuum AND aspiration)) AND (electrical OR electric OR manual) |
| Medication abortion | misoprostol OR methotrexate OR ethacridine OR rivanol OR nonsteroidal abortifacient agents OR nonsteroidal abortifacients |
| Later abortion in pregnancy/ weeks gestation | second trimester OR third trimester OR gestational age OR fetal viability OR foetal viability OR weeks of gestation OR late term abortion OR late-term abortion OR mid term abortion OR mid-term abortion |
| Social control/legal limit | social control OR formal social control OR control theory OR disciplinary infractions OR gatekeeping OR male domination OR social structure OR paternalism OR social engineering OR social norms OR social regulation OR legislation OR jurisprudence OR waiting period OR mandatory waiting OR restriction OR criminalization OR legal limit |

We added grey literature sources from scanning websites of research and non-governmental organizations that conduct abortion research (see full list in **S2 Table**) and additional hand selected articles from manual review of journals or reference lists of screened articles.

There is a lack of shared definitions and terminology regarding abortion at different stages in pregnancy [12]. This scoping review focuses on abortion later in pregnancy which we define as ≥12 weeks of gestation or labeled "second trimester". We also included studies of abortions at gestational legal limits that did not explicitly state weeks of gestation. We chose abortion later in pregnancy and gestational legal limits because most abortions occur early in pregnancy [13] and we wanted to capture circumstance of abortion at stages of abortion known to be associated with greater structural barriers and care seeking delays.

We included articles published in English, Spanish, French, and Portuguese that included description of the circumstance and/or socio-demographic or other characteristics of seeking or obtaining an induced abortion later in pregnancy and/or studies that reported the proportion of abortion occurring later in pregnancy. We also included studies on people who received care for complications following an induced abortion or denied an abortion later in pregnancy if circumstances for seeking an abortion were described.

We excluded studies that were narrative reviews; editorial articles; and randomized controlled trial or clinical intervention when the research design depends on a priori sample sizes for groups. Additionally, we excluded studies that were solely focused on spontaneous abortion and those that did not distinguish between induced and spontaneous abortion. We excluded studies that included only abortion for non-viable pregnancies or congenital anomalies. We chose to exclude these studies because abortion for congenital anomalies is well documented in the literature, more clearly understood, and less stigmatized than other circumstances [6, 14–16]. We also excluded conference abstracts because they are often preliminary, incomplete, or lack sufficient detail needed. See the population, intervention, control, outcome, timeframe, setting (PICOTS) criteria used in the scoping review in **Table 2**.

Author LJ conducted the title/abstract screen, full text review, and completed data extraction. LJ captured data on a standardized extraction form in Excel to collect information on country; study setting; data type; data year(s); sample size; weeks of gestation cutoff or range reported (at least ≥12 weeks when weeks of gestation were listed); proportion reported of

**Table 2. PICOTS criteria used in the scoping review.**

| PICOTS | |
|---|---|
| Population | Individuals who sought or obtain an induced abortion or post-abortion care for a complication of an induced abortion later in pregnancy. |
| Intervention | • Induced abortion later in pregnancy.<br>• Induced abortion later in pregnancy with complication.<br>• Receiving post abortion care later in pregnancy for complication from induced abortion<br>• Denied abortion later in pregnancy or due to gestational limits. |
| Control | None |
| Outcome | Qualitative of Quantitative data on:<br>• Description of the circumstance of seeking, obtaining, and/or receiving post abortion care for an induced abortion at later in pregnancy.<br>• Description of socio-demographic characteristics of individuals seeking, obtaining, and/or receiving post abortion care for an induced abortion at later in pregnancy.<br>• Studies that reported a population-based proportion of abortion occurring later in pregnancy.<br>• Studies that reported a facility-based proportion of abortion occurring later in pregnancy. |
| Timeframe | 1 January 2007–30 April 2024 |
| Setting | No limitations: all world regions, countries, states, communities |

abortion ≥12 weeks or labeled as second trimester; circumstance of a later abortion; associated socio-demographic characteristics; and any relevant limitations of the study. Studies were further categorized by use of qualitative data representing client accounts and use of quantitative data that describe sociodemographic or other characteristics associated with abortion later in pregnancy. To analyze the themes surrounding the circumstances from qualitative studies, we used a thematic analysis technique [17] to capture common themes that emerged. Data extraction and categorization of themes were conducted in the Excel form. An aim in this scoping review was to maximize the breadth of included studies. We did not assess the methodological quality of included studies.

## Results

The database searched yielded 2893 records after duplicates were removed (**Fig 1**).

We added 64 records from grey literature and removed 359 conference abstracts then screened 2598 records by title and/or abstract and of those included 735 records for full text review. Reasons articles were excluded upon full-text review include additional duplicates; duplicate data points; and the sample not meeting inclusion criteria (e.g., spontaneous abortion, non-viable pregnancy). We included a total of 78 studies. Of the included studies, 12 described the circumstances around seeking care for abortion later in pregnancy using qualitative data; 15 included quantitative data on sociodemographic or other characteristics associated with abortion later in pregnancy. Seventeen studies reported a population-based proportion of abortion later in pregnancy; and 45 reported a facility-based proportion of abortion later in pregnancy. Twelve studies included more than one type of key information. Results represented 30 countries. Details of included study characteristics are shown in **Table 3**.

### Circumstances of and sociodemographic characteristics associated with seeking care for abortion later in pregnancy

We identified 12 studies that described the circumstance around seeking care for abortion later in pregnancy from qualitative interviews and 15 studies that included quantitative information on sociodemographic or other characteristics associated with abortion later in pregnancy.

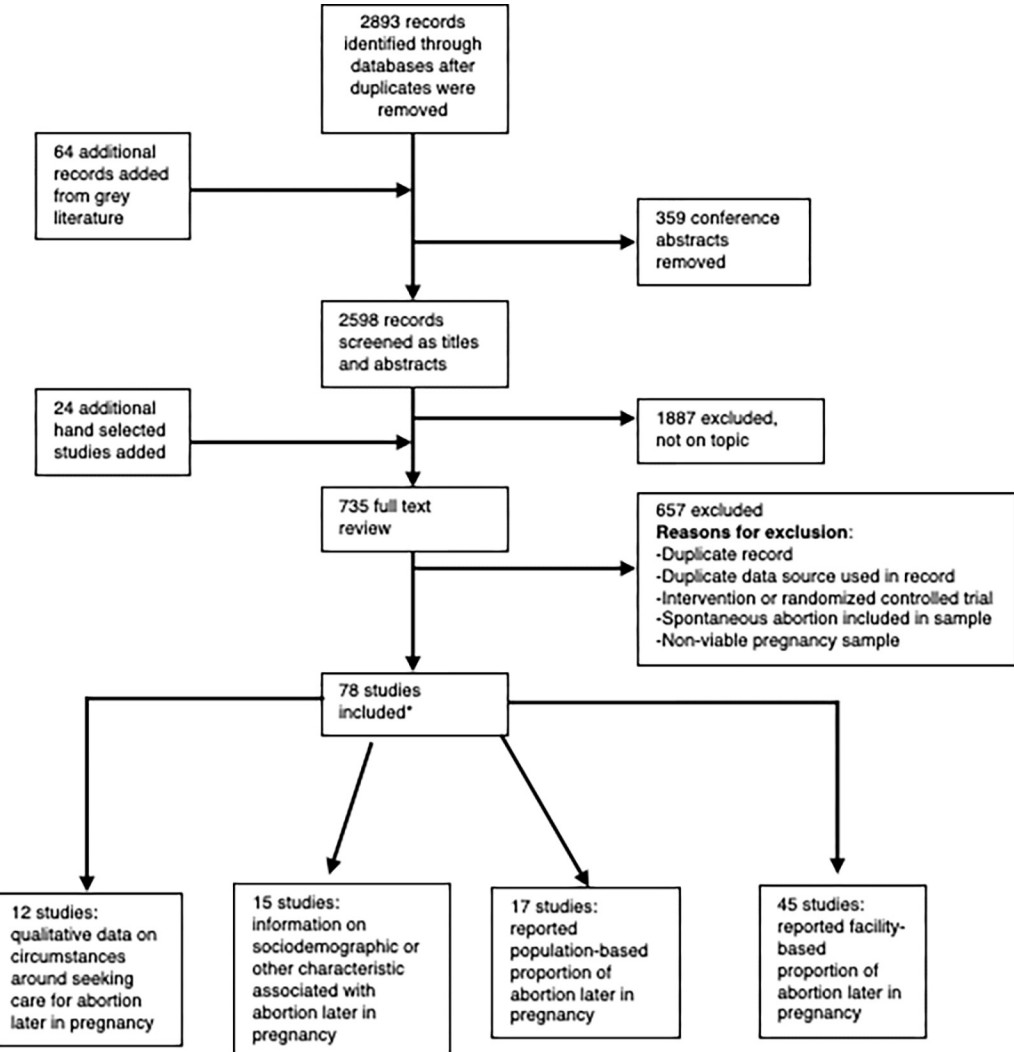

**Fig 1. PRISMA flow chart of included studies.** *Note that studies included do not add up to 78 because some had more than one type of included information (qualitative data on circumstances, sociodemographic or other characteristics, population, or facility-based proportion of abortion later in pregnancy).

From the 12 studies that reported qualitative data, we identified four main themes of circumstance of seeking an abortion later in pregnancy that occurred most frequently: health system challenges (12 studies), late pregnancy recognition (10 studies), financial challenges (8 studies), and delayed decision making (7 studies). These main themes as well as other circumstances identified from qualitative studies are detailed further in **Table 4A.**

Among the 15 studies that reported quantitative socio-demographic data or information on circumstances associated with abortion later in pregnancy, the most commonly reported characteristics and circumstances include low economic status or unemployment (10 studies); adolescence or young age (9 studies); health system challenges (8 studies); and late pregnancy recognition (7 studies). A full list of sociodemographic or other characteristics associated with seeking an abortion later in pregnancy, from quantitative data are shown in **Table 4B.**

Navigating health system obstacles was reported in all 12 of the analyses of qualitative data and eight of the studies with quantitative data (two studies contained both) [2, 5–7, 18, 19, 21,

**Table 3. Characteristics of included studies with reports of proportion and/or circumstance of abortion occurring at later in pregnancy.**

| Author, year | Country | Country legal status as described in the paper | Year of data | Type of data | Type of study sample/population | Sample size (interviews conducted or denominator of proportion) | Weeks of gestation cutoff or range reported (at least ≥12 weeks) | Proportion of abortion reported at specified weeks of gestation cutoff (y/n) | Included qualitative data on circumstances around seeking care for abortion ≥12 weeks (y/n) | Included information on sociodemographic or other characteristic associated with abortion ≥12 weeks |
|---|---|---|---|---|---|---|---|---|---|---|
| Gallo et al., 2007 [18] | Vietnam | Legal on request; second-trimester abortion restricted to specific facilities | 2005 | Qualitative interview | Purposive sample, induced abortion, 13–24 weeks | 60 | 13–24 | n | y | n |
| Harries et al., 2007 [19] | South Africa | Legal on request at ≤ 12 weeks | 2006 | Qualitative interview | Purposive sample, induced abortion, "second trimester" | 27 | 14–19.6 | n | y | n |
| Bagga et al., 2008 [20] | India | Legal up to 20 weeks | 1994–2006 | Medical records, 1 facility | Record review of all clients presenting for induced abortion | 3,096 | Second trimester, unspecified | y | n | n |
| Ingham et al., 2008 [21] | England & Wales | Legal up to 24 weeks; after only under specific conditions | 2005 | Facility administered survey of clients, 10 facilities | Purposive sample, induced abortion ≥13 weeks | 883 | ≥13 | n | n | y |
| Loeber & Wijsen 2008 [22] | The Netherlands | Legal on request up to 22 weeks when in a licensed facility by a physician | 2006 & 2008 | Administrative statistics, national registry | Population-based, induced abortion | 32,992 | 12–22 | y | n | y |
| Potdar et al., 2008 [23] | Cambodia | Legal on request up to 12 weeks, and after only under specific conditions | 2005 | Facility administered survey of clients, 5 facilities | Facility-based convenience sample, induced abortion | 110 | Self-reported second trimester | y | n | n |
| Rahim & Ara 2008 [24] | Pakistan | Legally restricted | 2000–2001 | Medical records, 1 facility | Facility-based convenience sample, induced abortion | 50 | Second trimester ≤16 weeks | y | n | n |
| Usta et al., 2008 [25] | Mozambique | Legal up to 12 weeks if the life or health of the women is threatened | 2005–2006 | Facility administered survey of clients, 5 facilities | Record review of all clients denied abortion for gestational limit | 1,734 | ≥13 | y | n | y |
| Bélanger & Oanh 2009 [26] | Vietnam | Legal on request | 2003 | Medical records, 1 facility | Facility-based sample, induced abortion with at least one child | 885 | "Second term" unspecified | y | n | n |
| Gebrehiwot & Liabsuetrakul, 2009 [27] | Ethiopia | Legal on request under specific conditions | 2003 & 2007 | Medical records, 6 facilities | Record review of all clients presenting for induced abortion with complication | 773 | >12 | y | n | n |

(*Continued*)

**Table 3.** (Continued)

| Author, year | Country | Country legal status as described in the paper | Year of data | Type of data | Type of study sample/ population | Sample size (interviews conducted or denominator of proportion) | Weeks of gestation cutoff or range reported (at least ≥12 weeks) | Proportion of abortion reported at specified weeks of gestation cutoff (y/n) | Included qualitative data on circumstances around seeking care for abortion ≥12 weeks (y/n) | Included information on sociodemographic or other characteristic associated with abortion ≥12 weeks |
|---|---|---|---|---|---|---|---|---|---|---|
| Rossier et al., 2009 [28] | France | Legal on request up to 12 weeks and 14 weeks under specific conditions | 2005 | Administrative statistics, national registry: facility reporting | Population-based, induced abortion | 91,607 (medical); 120,938 (surgical) | ≥12 | y | n | n |
| Kalyanwala et al., 2010 [29] | India | Legal under a wide range of conditions | 2007–2008 | Facility administered survey of clients, 16 facilities | Facility-based convenience sample, induced abortion, young unmarried women | 549 | Second trimester, unspecified | y | n | y |
| Kiley et al., 2010 [30] | USA | Not discussed | 2007–2008 | Medical records & facility survey, 1 facility | Facility-based convenience sample, induced abortion | 247 | 13–23 weeks & 3 days | y | n | y |
| Banerjee & Andersen 2012 [31] | India | Legal under a wide range of conditions | 2007 | Facility administered survey of clients, 10 facilities | Facility-based convenience sample, induced abortion with complication | 381 | 13–20 & >20 | y | n | n |
| Boersma et al., 2012 [32] | Curaçao | Completely prohibited by law | 2008–2009 | Administrative statistics, national registry | Population-based, induced abortion | 619 | >12 | y | n | n |
| Phaumvichit & Chandeying, 2012 [33] | Thailand | Legally restricted with exceptions for women's health or cases of rape | 2009–2010 | Medical records, 1 facility | Facility-based convenience sample, "illegal induced abortion" with complication | 84 | 14–20 & 22–28 | y | n | n |
| Ranji 2012, [34] | Iran | Legally restricted to therapeutic indications as defined by law | 2009–2010 | Facility administered survey of clients, 6 facilities | Facility-based convenience sample, induced abortion | 459 | >12 | y | n | n |
| Abiodun et al., 2013 [35] | Nigeria | Legally restricted except to save a women's life | 2005–2009 | Medical records, 1 facility | Record review of all clients presenting with complications of unsafe abortion | 96 | 13–20 & >20 | y | n | n |
| Foster & Kimport 2013 [5] | USA | Not discussed | 2008–2010 | Qualitative interview | Purposive sample, induced abortion ≥20 weeks | 272 | ≥20 | n | y | y |

*(Continued)*

**Table 3.** (Continued)

| Author, year | Country | Country legal status as described in the paper | Year of data | Type of data | Type of study sample/population | Sample size (interviews conducted or denominator of proportion) | Weeks of gestation cutoff or range reported (at least ≥12 weeks) | Proportion of abortion reported at specified weeks of gestation cutoff (y/n) | Included qualitative data on circumstances around seeking care for abortion ≥12 weeks (y/n) | Included information on sociodemographic or other characteristic associated with abortion ≥12 weeks |
|---|---|---|---|---|---|---|---|---|---|---|
| Grossman et al., 2013 [36] | USA | Not discussed | 2008–2010 | Administrative statistics, vital statistics and 15 facilities | Population-based, induced abortion | 9,054 | >13 | y | n | n |
| Kimport et al., 2013 [37] | USA | Not discussed | 2011 | Medical records, 1 facility | Record review of all clients presenting for induced abortion | 15,331 | 13–19 & ≥20 | y | n | n |
| Ojha & Bista 2013 [38] | Nepal | Legal as of 2002 | 2011–2012 | Medical records, 1 facility | Record review of all clients presenting for induced abortion with complication | 57 | 12–15 | y | n | n |
| Rocca et al., 2013 [39] | Nepal | Legal on request up to 12 weeks | 2010 | Medical records, 4 facilities | Facility-based sample, induced abortion with complication | 527 | ≥12 | y | n | n |
| Bonnen et al., 2014 [40] | Ethiopia | Legal up to 28 weeks under specific conditions | 2011–2012 | Administrative statistics, national registry | Population-based, induced abortion | 4,829 | Second trimester, unspecified | y | n | n |
| Dragoman et al., 2014 [41] | Multi country, 29 countries | Varying legal contexts | 2010 | Survey of facilities, 359 facilities | Facility-based sample, abortion-related severe maternal outcome | 295 | ≥14 | y | n | n |
| Janiak et al., 2014 [42] | USA | Not discussed | 2007–2009 | Medical records, 1 facility | Record review of all clients presenting for induced abortion at 19–24 weeks | 232 | 19–24 | n | n | y |
| Kouame et al., 2014 [43] | Cote d'Ivoire | Not discussed | 2006–2010 | Medical records, 3 facilities | Facility-based sample, induced abortion with complication | 1,982 | >12 | y | n | n |
| Mazuy et al., 2014 [44] | France | Legal up to 12 weeks | 2011 | Population-based survey | Population-based, induced abortion | 93,266 (surgical); 116,025 (medical) | >12 | y | n | n |

*(Continued)*

**Table 3.** (Continued)

| Author, year | Country | Country legal status as described in the paper | Year of data | Type of data | Type of study sample/ population | Sample size (interviews conducted or denominator of proportion) | Weeks of gestation cutoff or range reported (at least ≥12 weeks) | Proportion of abortion reported at specified weeks of gestation cutoff (y/n) | Included qualitative data on circumstances around seeking care for abortion ≥12 weeks (y/n) | Included information on sociodemographic or other characteristic associated with abortion ≥12 weeks |
|---|---|---|---|---|---|---|---|---|---|---|
| Nkwabong et al., 2014 [45] | Cameroon | Legally restricted with exceptions for cases of rape of incest | 2012 | Medical records, 2 facilities | Facility-based sample, induced abortion with complication | 94 | 14–22 | y | n | n |
| Prabhu, 2014 [46] | India | Not discussed | 2006–2010 | Facility administered survey of clients, 1 facility | Facility-based convenience sample, induced abortion, young unmarried women | 115 | Second trimester, unspecified | y | n | n |
| Purcell et al., 2014 [47] | Scotland | Abortion after 18 weeks is subject to significant limitations | 2013 | Qualitative interview | Purposive sample, induced abortion ≥16 weeks | 23 | ≥16 | n | y | n |
| Upadhyay et al., 2014 [48] | USA | Variable: the point of unspecified potential fetal viability established as the threshold after which states could restrict abortion care with exceptions to for the life and health of the pregnant woman | 2008–2010 | Qualitative interview & facility survey, 30 facilities | Purposive sample, induced abortion & denied abortion for gestational limit | 683 | ≥13 | n | y | y |
| Baum et al., 2015 [49] | Colombia | Legal under specific conditions | 2012 | Medical records, hospital system | Record review of all clients presenting for induced abortion | 200 | 12–15 | n | n | y |
| Mutua et al., 2015 [50] | Kenya | Legally restricted with exceptions only to save the women's life or health | 2012 | Survey of facilities, 350 facilities | Facility-based sample, induced abortion with complication | 2,631 | Second trimester, unspecified | y | n | n |
| Norman et al., 2015 [51] | Canada | Legal and federal legislation requires provincial and territorial health systems to provide abortion services | 2012 | Administrative statistics, 94 facilities | Population-based, induced abortion | 75,650 | Second trimester, unspecified | y | n | n |

*(Continued)*

**Table 3.** (Continued)

| Author, year | Country | Country legal status as described in the paper | Year of data | Type of data | Type of study sample/ population | Sample size (interviews conducted or denominator of proportion) | Weeks of gestation cutoff or range reported (at least ≥12 weeks) | Proportion of abortion reported at specified weeks of gestation cutoff (y/n) | Included qualitative data on circumstances around seeking care for abortion ≥12 weeks (y/n) | Included information on sociodemographic or other characteristic associated with abortion ≥12 weeks |
|---|---|---|---|---|---|---|---|---|---|---|
| Perry et al., 2015 [52] | USA | Not discussed | 2009–2013 | Medical records, 2 facilities | Facility-based sample, induced abortion resulting from rape | 19,465 | 14–23 weeks & 6 days | y | n | n |
| Puri et al., 2015 [53] | Nepal | Legal on request up to 12 weeks, up to 18 weeks if the pregnancy results from rape or incest, and at any time during pregnancy under specific conditions | 2013 | Qualitative interview | Purposive sample, denied abortion at >12 weeks | 25 | >12 | n | y | n |
| Upadhyay et al., 2015 [54] | USA | Not discussed | 2009–2010 | Medical claims data | Record review of induced abortions, Medi-Cal recipients | 54,911 | "after 12 weeks of gestation" | y | n | n |
| Erfani 2016 [55] | Iran | Legally restricted with exceptions for the life of the women or in cases of severe fetal abnormality | 2014 | Population-based survey | Population-based, induced abortion | 75 | 13–16 | y | n | n |
| French et al., 2016 [56] | USA | Nebraska has state restrictions: a 24-h waiting period, restrictions on insurance coverage and prohibition of telemedicine for abortion | 2014–2015 | Facility administered survey of clients, 3 facilities | Facility-based convenience sample, induced abortion | 353 | Second trimester, unspecified | y | n | n |
| Gerdts et al., 2016 [57] | Indonesia | Legally restricted with exceptions only in cases in which a woman's life is at risk or as the result of rape | 2012–2014 | Hotline records | Record review of all initial contacts to safe abortion hotline | 1,829 | >12 | y | n | n |
| Hossain et al., 2016 [58] | Bangladesh | Legally restricted with exceptions to save the woman's life; menstrual regulation services are permitted | 2014 | Qualitative interview | Purposive sample, denied abortion for gestational limit | 21 | >12 | n | y | n |

(Continued)

**Table 3.** (Continued)

| Author, year | Country | Country legal status as described in the paper | Year of data | Type of data | Type of study sample/ population | Sample size (interviews conducted or denominator of proportion) | Weeks of gestation cutoff or range reported (at least ≥12 weeks) | Proportion of abortion reported at specified weeks of gestation cutoff (y/n) | Included qualitative data on circumstances around seeking care for abortion ≥12 weeks (y/n) | Included information on sociodemographic or other characteristic associated with abortion ≥12 weeks |
|---|---|---|---|---|---|---|---|---|---|---|
| Karasek et al., 2016 [59] | USA | Arizona state restrictions: 24 hours waiting period | 2009–2010 | Facility administered survey of clients, 1 facility | Facility-based convenience sample, induced abortion | 326 | >14 | y | n | n |
| Kathpalia 2016 [60] | India | Abortion was legalized in India in 1971 | 2010–2014 | Medical records, 1 facility | Record review of all clients presenting for induced abortion with and without complication | 1,288 | Second trimester, unspecified | y | n | n |
| Madeiro & Diniz, 2016 [61] | Brazil | Legally restricted with exceptions for the life of the women, rape, and anencephaly of the fetus | 2013–2015 | Medical records, 68 facilities | Facility-based, induced abortion | 1,283 | 15–20 & >20 | y | n | n |
| Mossie Chekol et al., 2016 [62] | Ethiopia | Legal under specific conditions | 2014 | Facility administered survey of clients, 8 facilities | Facility-based convenience sample, induced abortion | 400 | Second trimester, unspecified | y | n | n |
| Blanchard et al., 2017 [63] | USA | At the time of this paper, nine states banned at 20 weeks, 32 states banned particular methods, and 27 states required providers to meet medically unnecessary regulations | 2012–2014 | Qualitative interview and facility administered survey of clients, 8 facilities | Purposive & convenience sample, induced abortion ≥14 weeks | 108 survey participants & 8 interviews | ≥14 | n | y | y |
| DePiñeres et al., 2017 [7] | Colombia | Authorized by the federal government, with no specific gestational age limitations, except that services after 15 weeks must be performed at a high level facility | 2013 | Qualitative interview | Purposive sample, denied abortion for gestational limit | 21 | 15–20 (1 at 30 weeks) | n | y | n |

(Continued)

**Table 3.** (Continued)

| Author, year | Country | Country legal status as described in the paper | Year of data | Type of data | Type of study sample/population | Sample size (interviews conducted or denominator of proportion) | Weeks of gestation cutoff or range reported (at least ≥12 weeks) | Proportion of abortion reported at specified weeks of gestation cutoff (y/n) | Included qualitative data on circumstances around seeking care for abortion ≥12 weeks (y/n) | Included information on sociodemographic or other characteristic associated with abortion ≥12 weeks |
|---|---|---|---|---|---|---|---|---|---|---|
| Ely et al., 2017 [64] | USA | State level restrictions vary | 2010–2015 | Case records | Record review of all induced abortion clients, abortion funds recipients | 3,999 | Second trimester, unspecified | y | n | n |
| Gerdts et al., 2017 [65] | South Africa | Legal on request up to 12 weeks, and for socioeconomic or medical reasons from 12 to 20 weeks. Beyond 20 weeks, permission of two medical practitioners is required | 2015 | Survey of clients, informal sector | Respondent driven sample, induced abortion, self-managed | 42 | ≥12 | y | n | n |
| Johns et al., 2017 [66] | USA | Not discussed | 2011–2012 | Claims data | Record review of all induced abortion clients, Medi-Cal recipients | 35,431 | Second trimester, unspecified | y | n | n |
| Jones & Jerman, 2017 [67] | USA | Nationally legalized in 1973 | 2014–2015 | Survey of facilities, 87 facilities | Population-based, induced abortion | 8,380 | ≥13 | y | n | y |
| Madeiro & Rufino, 2017 [68] | Brazil | Not discussed | 2012–2013 | Facility administered survey of clients, 1 facility | Facility-based sample, induced abortion with complication | 78 | ≥13 | y | n | n |
| Shankar et al., 2017 [69] | Australia | Not discussed | 2014–2015 | Facility administered survey of clients, 14 facilities | Facility-based convenience sample, induced abortion | 2,129 | Second trimester, unspecified | y | n | n |
| Saavedra-Avendano et al., 2018 [70] | Mexico | Abortion law is determined at the state level; first trimester abortion was decriminalized in Mexico City | 2014 | Medical records, 4 facilities | Record review of all clients-compared denied abortion for gestational limit vs. not | 52,391 | ≥12 | n | n | y |
| Ushie et al., 2018 [71] | Kenya | The Kenyan Constitution offers potential for increasing women's access to safe abortion | 2012 | Facility administered survey of clients, 328 facilities | Facility-based sample, induced abortion with complication, adolescents | 398 | >12 | y | n | n |

*(Continued)*

Table 3. (Continued)

| Author, year | Country | Country legal status as described in the paper | Year of data | Type of data | Type of study sample/population | Sample size (interviews conducted or denominator of proportion) | Weeks of gestation cutoff or range reported (at least ≥12 weeks) | Proportion of abortion reported at specified weeks of gestation cutoff (y/n) | Included qualitative data on circumstances around seeking care for abortion ≥12 weeks (y/n) | Included information on sociodemographic or other characteristic associated with abortion ≥12 weeks |
|---|---|---|---|---|---|---|---|---|---|---|
| Williams et al., 2018 [72] | USA | Arizona's state legislature has enacted multiple laws restricting the provision of abortion | 2012–2013 | Administrative statistics | Population-based, induced abortion | 26,338 | ≥14 | y | n | n |
| Dasgupta et al., 2019 [73] | India | Legal since the medical termination of pregnancy Act, 1971 | 2015–2016 | Community-based survey | Population-based, induced abortion | 86 | >12 | y | n | n |
| Jones et al., 2019 [74] | USA | Legal restrictions vary at the state level | 2017 | Administrative & facility statistics | Population-based, induced abortion | NA | ≥13 | y | n | n |
| Popinchalk & Sedgh 2019 [4] | Multi country, 24 countries | Varying legal contexts | 2017 or most recent available | Administrative & facility statistics | Population-based, induced abortion | NA | Variable between countries, but reported as ≥13 in the paper | y | n | n |
| Restrepo-Bernal et al., 2019 [75] | Colombia | Legal under specific conditions | 2013–2014 | Medical records, 1 facility | Record review of all clients presenting for induced abortion | 87 | 18–26 | y | n | n |
| Van de Velde et al., 2019 [76] | Belgium | Legal on request up to 14 weeks with 6 day waiting period | 2013–2016 | Medical records, hospital system | Record review of all clients presenting for induced abortion | 28,741 | Two categories: 1. between 13 weeks + 2 days and 13 weeks + 6 days. 2. ≥14 2 | y | n | y |
| Fuentes et al., 2020 [77] | USA | Texas has implemented a series of laws restricting access | 2012 & 2014 | Facility administered survey of clients, 8 facilities | Facility-based convenience sample, induced abortion | 721 | 13–15 & ≥16 | y | n | n |
| Goyal et al., 2020 [78] | USA | In 2013, the Texas state legislature passed House Bill 2, a restrictive abortion law | 2015–2016 | Medical records, 8 facilities | Record review of all clients presenting for induced abortion | 24,555 | 12–14; 15–17; & 18–24 weeks | y | n | y |

(Continued)

**Table 3.** (Continued)

| Author, year | Country | Country legal status as described in the paper | Year of data | Type of data | Type of study sample/ population | Sample size (interviews conducted or denominator of proportion) | Weeks of gestation cutoff or range reported (at least ≥12 weeks) | Proportion of abortion reported at specified weeks of gestation cutoff (y/n) | Included qualitative data on circumstances around seeking care for abortion ≥12 weeks (y/n) | Included information on sociodemographic or other characteristic associated with abortion ≥12 weeks |
|---|---|---|---|---|---|---|---|---|---|---|
| Kebede et al., 2020 [79] | Ethiopia | Liberalized abortion law since 2005 | 2019–2020 | Facility administered survey of clients, 1 facility | Facility-based convenience sample, induced abortion | 238 | 13–20 & >20 | y | n | y |
| Shapiro et al., 2020 [80] | USA | Varies at the state level | 2012–2016 | Administrative statistics, vital statistics | Population-based, induced abortion | 137,128 | ≥20 | y | n | n |
| Sharma & Pradhan 2020 [81] | India | Legal for a broad range of medical and social reasons | 2015–2016 | Population-based survey | Population-based, induced abortion | 6,876 | ≥20 | y | n | n |
| De Zordo et al., 2021 [2] | UK, Netherlands, Spain | Legal upon request, or on broad social or economic grounds, in nearly all European countries, with variability on regulatory and procedural barriers | 2017–2019 | Qualitative interview | Purposive sample, induced abortion, needed to travel | 30 | Traveling because of gestational limit | n | y | n |
| Moseson et al., 2021 [82] | USA | Not discussed | 2019 | Online survey | Purposive sample, induced abortion, trans and gender expansive people | 67 | 13–15, 16–20, & 21–24 | y | n | n |
| Mouba et al., 2021 [83] | Gabon | Legally restricted | 2014–2018 | Medical records, 1 facility | Facility-based sample, induced abortion, clandestine with complication | 128 | ≥12 | y | n | n |
| Kimport 2022 [6] | USA | Varies at the state level | 2018 | Qualitative interview | Purposive sample, induced abortion ≥24 weeks | 28 | 24–25 | n | y | n |
| Schummers et al., 2022 [84] | Canada | Legal | 2017–2020 | Administrative statistics | Population-based, induced abortion | 84,032 | Second trimester, ≥14 | y | n | n |
| Trapani et al., 2022 [85] | Brazil | Legally restricted with exception for cases of rape | 2014–202 | Medical records, 1 facility | Facility-based convenience sample, care seeking for induced abortion after sexual violence | 178 | 12–16, 16–20, >20 | y | n | n |

*(Continued)*

Table 3. (Continued)

| Author, year | Country | Country legal status as described in the paper | Year of data | Type of data | Type of study sample/ population | Sample size (interviews conducted or denominator of proportion) | Weeks of gestation cutoff or range reported (at least ≥12 weeks) | Proportion of abortion reported at specified weeks of gestation cutoff (y/n) | Included qualitative data on circumstances around seeking care for abortion ≥12 weeks (y/n) | Included information on sociodemographic or other characteristic associated with abortion ≥12 weeks |
|---|---|---|---|---|---|---|---|---|---|---|
| White et al., 2022 [86] | USA | Mississippi has one of the most restrictive abortion policy environments in the United States | 2018 | Facility statistics, 12 facilities | Facility-based convenience sample, induced abortion | 4,455 | 12–15, ≥16 | y | n | n |
| Gonzolez-Perez et al., 2023 [87] | Colombia | Colombia decriminalized induced abortion in 2006 under specific conditions | 2015–2021 | Medical records, 5 facilities | Facility-based convenience sample, induced abortion | 20,423 | ≥15 | y | n | n |
| Jubert et al., 2023 [88] | The Netherlands | In Europe, the legal time limit to perform an abortion varies from 12 weeks in Portugal/France to 24 weeks in the Netherlands | 2020 | Facility administered survey of clients, 1 facility | Purposive sample, induced abortion, needed to travel from France | 35 | Traveling because of gestational limit, >16 reported | y | n | n |
| Makleff et al., 2023 [89] | USA | Varies at the state level and increasing in restrictiveness | 2020–2021 | Qualitative interview | Purposive sample, traveled for later abortion | 19 | 13–24 | n | y | n |
| Malik et al., 2023 [90] | India | Legal under various circumstances since the enactment of the Medical Termination of Pregnancy (MTP) Act in the 1970s | 2019–2021 | Population-based survey | Population-based, induced abortion | 665,671 | Second or third trimester, unspecified | y | n | n |

**Table 4. Circumstances of and sociodemographic or other characteristics associated with seeking care for abortion at later in pregnancy.**

**A. Circumstance of seeking care for an abortion later in pregnancy from qualitative data**

| Category | Themes and issues presented | First author, year, reference number |
|---|---|---|
| Health system challenges | Negative and judgmental attitudes from staff; inappropriate referrals; not knowing where to find provider; not knowing what service are available to them; needing to travel; insurance problems; unaware right to access care withing sanctioned period; lack of information where to get the abortion service; scarcity of accessible abortion providers in their area of residence; gestational age miscalculated by health professionals; needing time to coordinate the trip to the hospital | Gallo 2007 [18]; Harries 2007 [19]; Foster 2013 [5]; Purcell 2014 [47]; Upadhyay 2014 [48]; Puri 2015 [53]; Hossain 2016 [58]; Blanchard 2017 [63]; Depiñeres 2017 [7]; De Zordo 2021 [2]; Kimport 2022 [6]; Makleff 2023 [89] |
| Late pregnancy recognition | Irregular menses; contraceptive use; lack of pregnancy symptoms; unawareness of pregnancy symptoms; misinformation given by or misunderstandings with health professionals about contraception | Gallo 2007 [18]; Harries 2007 [19]; Foster 2013 [5]; Purcell 2014 [47]; Upadhyay 2014 [48]; Puri 2015 [53]; Hossain 2016 [58]; Depiñeres 2017 [7]; De Zordo 2021 [2]; Kimport 2022 [6] |
| Financial challenges | Having to raise money for travel and procedure costs | Gallo 2007 [18]; Foster 2013 [5]; Upadhyay 2014 [48]; Puri 2015 [53]; Hossain 2016 [58]; Blanchard 2017 [63]; Depiñeres 2017 [7]; Kimport 2022 [6] |
| Delayed decision-making | Ambivalence; uncertainty; fear of other's reactions; weighing many factors (ie concern for existing children, finances, expectations from family); need for time to decide | Gallo 2007 [18]; Harries 2007 [19]; Purcell 2014 [47]; Puri 2015 [53]; Hossain 2016 [58]; Depiñeres 2017 [7]; Makleff 2023 [89] |
| Other Logistics | difficulty in getting time off work; difficulty in getting a driver; care for other children | Puri 2015 [53]; Hossain 2016 [58]; Makleff 2023 [89] |
| Relationship difficulties | No stable partner; partner left or was aggressive | Gallo 2007 [18]; Foster 2013 [5] |
| Own or family health concerns | | Puri 2015 [53]; Hossain 2016 [58] |
| Sex of the fetus | | Puri 2015 [53]; Hossain 2016 [58] |
| Policy related barriers | | Kimport 2022 [6]; Makleff 2023 [89] |
| Belief that infrequent intercourse made pregnancy unlikely | | Gallo 2007 [18] |
| Shame & fear | | Harries 2007 [19] |
| New observed serious fetal health issue | | Kimport 2022 [6] |

**B. Sociodemographic or other characteristics associated with seeking an abortion later in pregnancy, from quantitative data**

| Category | Themes and issues presented | First author, year, reference number |
|---|---|---|
| Low economic status or unemployment | | Usta 2008, [25]; Loeber 2008 [22]; Kiley 2010 [30]; Janiak 2014 [42]; Upadhyay 2014 [48]; Baum 2015 [49]; Blanchard 2017 [63]; Jones 2017 [67]; Van de Velde 2019 [76]; Goyal 2020 [78] |
| Adolescent or young age | | Loeber 2008 [22]; Usta 2008 [25]; Kalyanwala 2010 [29]; Kiley 2010 [30]; Foster 2013 [5]; Upadhyay 2014 [48]; Jones 2017 [67]; Saavedra-Avendano 2017 [70]; Van de Velde 2019 [76] |

*(Continued)*

**Table 4.** (Continued)

### A. Circumstance of seeking care for an abortion later in pregnancy from qualitative data

| Category | Themes and issues presented | First author, year, reference number |
|---|---|---|
| Health system challenges | Insurance status; not knowing where to find provider | Ingham 2008 [21]; Loeber 2008 [22]; Kiley 2010 [30]; Upadhyay 2014 [48]; Baum 2015 [49]; Blanchard 2017 [63]; Jones 2017 [67]; Kebede 2020 [79]; |
| Late pregnancy recognition | Current contraceptive use | Ingham 2008 [21]; Loeber 2008 [22];; Janiak 2014 [42]; Upadhyay 2014 [48]; Baum 2015 [49]; Jones 2017 [67]; Kebede 2020 [79] |
| Living far from abortion services | | Loeber 2008 [22]; Kiley 2010 [30]; Baum 2015 [49]; Jones 2017 [67]; Saavedra-Avendano 2017 [70]; Goyal 2020 [78] |
| Low/less education | | Usta 2008, [25]; Jones 2017 [67]; Saavedra-Avendano 2017 [70]; Van de Velde 2019 [76] |
| Delayed decision-making | Ambivalence | Ingham 2008 [21]; Loeber 2008 [22]; Janiak 2014 [42]; Kebede 2020 [79] |
| Experienced gender-based violence | | Kalyanwala 2010 [29]; Foster 2013 [5] |
| Black race (USA) | | Jones 2017 [67]; Goyal 2020 [78] |
| Immigration status | | Loeber 2008 [22]; Van de Velde 2019 [76] |
| Parity | Too early to have kids; family was complete; spacing children | Loeber 2008 [22] |
| Older maternal age | | Loeber 2008 [22] |
| Special needs education | | Van de Velde 2019 [76] |
| Other Logistics | Difficulty in getting time off work, difficulty in getting a driver, care for other | Baum 2015 [49] |
| Relationship difficulties | | Loeber 2008 [22] |
| Underestimated own gestational age | | Janiak 2014 [42] |

22, 47–49, 53, 58, 63, 67, 79, 89]. These health system challenges included several issues including lack of information on where and how to access care and limited provider availability (**see Table 4 for full list).** Late pregnancy recognition also emerged as a common theme related to seeking an abortion later in pregnancy, identified in ten qualitative and seven quantitative studies (one study contained both) [2, 5–7, 18, 19, 21, 22, 42, 47–49, 53, 58, 63, 67, 79]. Further explanation for late pregnancy recognition included irregular menses; current contraceptive use; lack of pregnancy symptoms; unawareness of pregnancy symptoms; misinformation given by or misunderstandings with health professionals about contraception (**Table 4**). These circumstances led women to wait until the second or third month of a missed period before seeking pregnancy confirmation which contributed to delays in seeking care [19]. Similarly, participants of a different study reported monthly bleeding that they attributed to menses, leading them to believe they were not pregnant [47].

Participants in eight qualitative studies reported financial challenges that contributed to delays in their ability to access abortion care until later in pregnancy [5–7, 18, 48, 53, 58, 63]. Similarly, low economic status or unemployment was associated with seeking abortion care later in 10 studies with quantitative data [22, 25, 30, 42, 48, 49, 63, 67, 76, 78]. These challenges included needing time to raise money for the procedure and for travel in some cases.

Decision delays were part of the circumstances for needing an abortion later in pregnancy in seven studies with qualitative data and four with quantitative data [7, 18, 19, 21, 22, 42, 47, 53, 58, 79, 89]. These decision-making delays were described as ambivalence; uncertainty; needing time to decide; and weighing many competing and intersecting factors when making their decisions such as concerns for the health and well-being of existing children, financial constraints, expectations from family members, their current gestational age, and the desire to carry a pregnancy to term. Decision-making delays intersected with health system barriers and stigma to compound and extend service delays [5, 6, 18, 19, 48, 63, 89] (**Table 4**).

## Reports of population and facility-based proportion of abortions later in pregnancy

Seventeen studies contained reports of population-based proportion of abortion later in pregnancy ($\geq$12 weeks of gestation or labeled as second trimester) from national, subnational, and community-based data sources. These studies included data from eight countries and reported proportions of abortions later in pregnancy that ranged from 2–33% of all abortions [4, 80]. There was a lack of standardized categorization when reporting duration of gestation and known methodological limitations to direct reporting of abortion in surveys [91, 92], which partially explains this wide range.

Forty-five studies contained facility-based reports of proportion of abortions later in pregnancy. These studies reported varying populations, varying number of facilities, different denominators, and non-standardized categories for abortion occurring later in pregnancy. Studies with a reported proportion of abortion later in pregnancy are listed with any additional context explaining the reported proportion in **Table 5.** These descriptions include information on the settings, populations, care facility, or other factors.

## Discussion

We provide a comprehensive scoping review of existing literature that reports circumstances of abortion later in pregnancy, associated sociodemographic or other characteristics, and/or reports of population and facility-based proportion of induced abortion occurring later in pregnancy. Overall, we show a dearth of studies in this area; with only 30 countries represented and some countries over-represented. Our findings revealed themes across settings such as health system challenges, delayed pregnancy recognition, financial challenges, and delayed decision-making that contributed to the circumstances of abortion later in pregnancy. Reports of proportion of abortions occurring later in pregnancy lack standard definitions, terminology, and measures for abortion later in pregnancy.

These findings elucidate the contexts and circumstances in which individuals need later abortions and provide insight into policy and health system solutions. Further, young people and those with lower levels of income were over-represented among those who need abortion later in pregnancy. Universal access to sexual and reproductive health information and services is a key strategy towards improving sexual and reproductive outcomes and ensuring individual and population health and respect for human rights [3] and will contribute to progress towards Sustainable Development Goals [94]. People have and will continue to need abortion later in pregnancy, so it is important to understand patterns in circumstances surrounding seeking care.

Studies that report the circumstances surrounding seeking an abortion later in pregnancy illustrate that people are seeking and accessing abortion care later for overlapping and intersecting reasons that compound and extend delays. Late pregnancy recognition puts people at a disadvantage for accessing care early, and financial and health system challenges accessing

**Table 5. Context and explanation for reported proportion of abortion later in pregnancy.**

| Author, year | Country | Year of data | Type of data | Type of study sample/population | Sample size (interviews conducted or denominator of proportion) | Weeks of gestation cutoff or range reported (at least ≥12 weeks) | Proportion of abortion later in pregnancy | Context and explanation for reported proportion of abortion later in pregnancy |
|---|---|---|---|---|---|---|---|---|
| Bagga et al., 2008 [20] | India | 1994–2006 | Medical records, 1 facility | Record review of all clients presenting for induced abortion | 3,096 | Second trimester, unspecified | 18.2 | NA |
| Loeber & Wijsen 2008 [22] | The Netherlands | 2006 & 2008 | Administrative statistics, national registry | Population-based, induced abortion | 32,992 | 12–22 | 6.6 | This paper notes that some hospitals do not report abortions due to antenatal diagnosis stating that this trajectory is different from abortion of social grounds. Other hospitals make distinctions based on method (ie curettage is within abortion framework of law while induction of labor is reported as premature birth) resulting in inconsistent reporting |
| Potdar et al., 2008 [23] | Cambodia | 2005 | Facility administered survey of clients, 5 facilities | Facility-based convenience sample, induced abortion | 110 | Self-reported second trimester | 4.6 | NA |
| Rahim & Ara 2008 [24] | Pakistan | 2000–2001 | Medical records, 1 facility | Facility-based convenience sample, induced abortion | 50 | Second trimester ≤16 weeks | 8 | NA |
| Usta et al., 2008 [25] | Mozambique | 2005–2006 | Facility administered survey of clients, 5 facilities | Record review of all clients denied abortion for gestational limit | 1,734 | ≥13 | 4.5 | NA |
| Bélanger & Oanh 2009 [26] | Vietnam | 2003 | Medical records, 1 facility | Facility-based sample, induced abortion with at least one child | 885 | "Second term" unspecified | 14.6 | This study suggests that some women in the sample use second term abortions as a means of sex selection by demonstrating that sonless women with at least two living daughters are more likely to undergo a second-term abortion than a first-term abortion. |
| Gebrehiwot & Liabsuetrakul, 2009 [27] | Ethiopia | 2003 & 2007 | Medical records, 6 facilities | Record review of all clients presenting for induced abortion with complication | 773 | >12 | 40.8 | These data were collected from Tikur Anbessa Hospital (TAH), in Addis Ababa, a tertiary referral hospital with medical and surgical intensive care units handling nearly all complicated obstetrics and gynecologic cases, therefore a higher proportion of abortions with complications are seen there, relative to other facilities in the country. |

(*Continued*)

**Table 5.** (Continued)

| Author, year | Country | Year of data | Type of data | Type of study sample/ population | Sample size (interviews conducted or denominator of proportion) | Weeks of gestation cutoff or range reported (at least ≥12 weeks) | Proportion of abortion later in pregnancy | Context and explanation for reported proportion of abortion later in pregnancy |
|---|---|---|---|---|---|---|---|---|
| Rossier et al., 2009 [28] | France | 2005 | Administrative statistics, national registry: facility reporting | Population-based, induced abortion | 91,607 (medical); 120,938 (surgical) | ≥12 | 2.5 & 15.3 respectively | NA |
| Kalyanwala et al., 2010 [29] | India | 2007–2008 | Facility administered survey of clients, 16 facilities | Facility-based convenience sample, induced abortion, young unmarried women | 549 | Second trimester, unspecified | 25.3 | This study sample was constructed from young, unmarried women who are less likely to recognize pregnancy early and more likely to experience obstacles obtaining care. |
| Kiley et al., 2010 [30] | USA | 2007–2008 | Medical records & facility survey, 1 facility | Facility-based convenience sample, induced abortion | 247 | 13–23 weeks & 3 days | 32.0 | NA |
| Banerjee & Andersen 2012 [31] | India | 2007 | Facility administered survey of clients, 10 facilities | Facility-based sample, induced abortion with complication | 381 | 13–20 & >20 | 6 | This study sought to recruit women only with post abortion complications |
| Boersma et al., 2012 [32] | Curaçao | 2008–2009 | Administrative statistics, national registry | Population-based, induced abortion | 619 | >12 | 5.4 | This paper notes that more than half of the pregnancies were terminated before the seventh week. The authors interpreted this as "a high awareness of women to detect their pregnancy" represented in these data |
| Phaumvichit & Chandeying, 2012 [33] | Thailand | 2009–2010 | Medical records, 1 facility | Facility-based convenience sample, "illegal induced abortion" with complication | 84 | 14–20 & 22–28 | 17.9 & 5.9 respectively | This study was conducted among women admitted to the hospital with illegal induced abortion |
| Ranji 2012, [34] | Iran | 2009–2010 | Facility administered survey of clients, 6 facilities | Facility-based convenience sample, induced abortion | 459 | >12 | 12.9 | NA |
| Abiodun et al., 2013 [35] | Nigeria | 2005–2009 | Medical records, 1 facility | Record review of all clients presenting with complications of unsafe abortion | 96 | 13–20 & >20 | 41.7 & 6.2 respectively | The study sampled women treated for complications of unsafe abortion. The authors explain that the overwhelming majority of the patients in this study were young unmarried students among whom pregnancy is viewed with strong social disapproval which may present additional barriers to care |
| Grossman et al., 2013 [36] | USA | 2008–2010 | Administrative statistics, vital statistics and 15 facilities | Population-based, induced abortion | 9,054 | >13 | 3.5 | NA |

*(Continued)*

**Table 5.** (Continued)

| Author, year | Country | Year of data | Type of data | Type of study sample/ population | Sample size (interviews conducted or denominator of proportion) | Weeks of gestation cutoff or range reported (at least ≥12 weeks) | Proportion of abortion later in pregnancy | Context and explanation for reported proportion of abortion later in pregnancy |
|---|---|---|---|---|---|---|---|---|
| Kimport et al., 2013 [37] | USA | 2011 | Medical records, 1 facility | Record review of all clients presenting for induced abortion | 15,331 | 13–19 & ≥20 | 10.4 &2.9 respectively | NA |
| Ojha & Bista 2013 [38] | Nepal | 2011–2012 | Medical records, 1 facility | Record review of all clients presenting for induced abortion with complication | 57 | 12–15 | 8.7 | This study included a small sample of women presenting to hospital with complications |
| Rocca et al., 2013 [39] | Nepal | 2010 | Medical records, 4 facilities | Facility-based sample, induced abortion with complication | 527 | ≥12 | 10.3 | This sample was of women presenting to hospital with complications |
| Bonnen et al., 2014 [40] | Ethiopia | 2011–2012 | Administrative statistics, national registry | Population-based, induced abortion | 4,829 | Second trimester, unspecified | 4.0 | NA |
| Dragoman et al., 2014 [41] | Multi country, 29 countries | 2010 | Survey of facilities, 359 facilities | Facility-based sample, abortion–related severe maternal outcome | 295 | ≥14 | 32.9 | This sample was of women with severe abortion-related maternal outcomes |
| Kouame et al., 2014 [43] | Cote d'Ivoire | 2006–2010 | Medical records, 3 facilities | Facility-based sample, induced abortion with complication | 1,982 | >12 | 31.6 | This sample was of women with complications of illegal induced abortions admitted to an intensive care unit |
| Mazuy et al., 2014 [44] | France | 2011 | Population-based survey | Population-based, induced abortion | 93,266 (surgical); 116,025 (medical) | >12 | 15.7 (surgical); 1.5 (medical) | NA |
| Nkwabong et al., 2014 [45] | Cameroon | 2012 | Medical records, 2 facilities | Facility-based sample, induced abortion with complication | 94 | 14–22 | 21.3 | The sample was of women with out of facility clandestine abortions but who were seen in a facility for complications |
| Prabhu, 2014 [46] | India | 2006–2010 | Facility administered survey of clients, 1 facility | Facility-based convenience sample, induced abortion, young unmarried women | 115 | Second trimester, unspecified | 72.2 | This sample was of young and unmarried women who are less likely than most abortion clients to recognize early pregnancy and have resources to access care |
| Mutua et al., 2015 [50] | Kenya | 2012 | Survey of facilities, 350 facilities | Facility-based sample, induced abortion with complication | 2,631 | Second trimester, unspecified | 37.0 | The sample was of women seeking post abortion care in healthcare facilities. This study had a large number of facilities and the authors noted that referrals were a source of delay that could result in more abortions later in pregnancy represented in this sample. |
| Norman et al., 2015 [51] | Canada | 2012 | Administrative statistics, 94 facilities | Population-based, induced abortion | 75,650 | Second trimester, unspecified | 6.3 | NA |

(*Continued*)

**Table 5.** (Continued)

| Author, year | Country | Year of data | Type of data | Type of study sample/ population | Sample size (interviews conducted or denominator of proportion) | Weeks of gestation cutoff or range reported (at least ≥12 weeks) | Proportion of abortion later in pregnancy | Context and explanation for reported proportion of abortion later in pregnancy |
|---|---|---|---|---|---|---|---|---|
| Perry et al., 2015 [52] | USA | 2009–2013 | Medical records, 2 facilities | Facility-based sample, induced abortion resulting from rape | 19,465 | 14–23 weeks + 6 days | 2.4 | This sample was of women seeking abortion resulting from rape. The authors explain that reasons why pregnancies may present later for termination may be due to late pregnancy recognition |
| Upadhyay et al., 2015 [54] | USA | 2009–2010 | Claims data | Record review of induced abortions, Medi-Cal recipients | 54,911 | "after 12 weeks of gestation" | 16.1 | This sample was of Medicaid recipients and may not be representative of all who seek abortion care. Medicaid recipients are more likely to possess some characteristics associated with abortion later in pregnancy (ie lower levels of income) |
| Erfani 2016 [55] | Iran | 2014 | Population-based survey | Population-based, induced abortion | 75 | 13–16 | 6.7 | NA |
| French et al., 2016 [56] | USA | 2014–2015 | Facility administered survey of clients, 3 facilities | Facility-based convenience sample, induced abortion | 353 | Second trimester, unspecified | 9.0 | NA |
| Gerdts et al., 2016 [57] | Indonesia | 2012–2014 | Hotline records | Record review of all initial contacts to safe abortion hotline | 1,829 | >12 | 18.3 | This sample was of initial contacts to safe abortion hotline, it's not known what proportion received an abortion |
| Karasek et al., 2016 [59] | USA | 2009–2010 | Facility administered survey of clients, 1 facility | Facility-based convenience sample, induced abortion | 326 | >14 | 3.1 | NA |
| Kathpalia 2016 [60] | India | 2010–2014 | Medical records, 1 facility | Record review of all clients presenting for induced abortion with and without complication | 1,288 | Second trimester, unspecified | 5.5 | NA |
| Madeiro & Diniz, 2016 [61] | Brazil | 2013–2015 | Medical records, 68 facilities | Facility-based, induced abortion | 1,283 | 15–20 & >20 | 27.0 & 5.0 respectively | NA |
| Mossie Chekol et al., 2016 [62] | Ethiopia | 2014 | Facility administered survey of clients, 8 facilities | Facility-based convenience sample, induced abortion | 400 | Second trimester, unspecified | 11.0 | The primary purpose of this study was to identify factors associated with women's satisfaction with comprehensive abortion care and the sampled facilities were chosen based on availability of comprehensive abortion care services and high caseloads. Therefore, they may not be representative of all facilities. |

(*Continued*)

**Table 5.** (Continued)

| Author, year | Country | Year of data | Type of data | Type of study sample/population | Sample size (interviews conducted or denominator of proportion) | Weeks of gestation cutoff or range reported (at least ≥12 weeks) | Proportion of abortion later in pregnancy | Context and explanation for reported proportion of abortion later in pregnancy |
|---|---|---|---|---|---|---|---|---|
| Ely et al., 2017 [64] | USA | 2010–2015 | Case records | Record review of all induced abortion clients, abortion funds recipients | 3,999 | Second trimester, unspecified | 74.8 | This sample was of all abortion fund recipient who by nature of accessing abortion fund resources were disadvantaged for seeking timely care |
| Gerdts et al., 2017 [65] | South Africa | 2015 | Survey of clients, informal sector | Respondent driven sample, induced abortion, self-managed | 42 | ≥12 | 21.0 | This sample was of women seeking abortion in the informal sector and may not be generalizable to a broader population |
| Johns et al., 2017 [66] | USA | 2011–2012 | Claims data | Record review of all induced abortion clients, Medi-Cal recipients | 35,431 | Second trimester, unspecified | 15.7 | This sample was Medicaid recipients and may not be representative of all who seek abortion care. Medicaid recipients are more likely to possess some characteristics associated with abortion later in pregnancy (ie lower levels of income) |
| Jones & Jerman, 2017 [67] | USA | 2014–2015 | Survey of facilities, 87 facilities | Population-based, induced abortion | 8,380 | ≥13 | 10.0 | NA |
| Madeiro & Rufino, 2017 [68] | Brazil | 2012–2013 | Facility administered survey of clients, 1 facility | Facility-based sample, induced abortion with complication | 78 | ≥13 | 15.4 | This was a small sample of women presenting to a hospital with complications |
| Shankar et al., 2017 [69] | Australia | 2014–2015 | Facility administered survey of clients, 14 facilities | Facility-based convenience sample, induced abortion | 2,129 | Second trimester, unspecified | 3.7 | NA |
| Ushie et al., 2018 [71] | Kenya | 2012 | Facility administered survey of clients, 328 facilities | Facility-based sample, induced abortion with complication, adolescents | 398 | >12 | 42.5 | This sample was adolescents and young women presenting to facilities with complications |
| Williams et al., 2018 [72] | USA | 2012–2013 | Administrative statistics | Population-based, induced abortion | 26,338 | ≥14 | 8.7 | NA |
| Dasgupta et al., 2019 [73] | India | 2015–2016 | Community-based survey | Population-based, induced abortion | 86 | >12 | 8.1 | NA |
| Jones et al., 2019 [74] | USA | 2017 | Administrative & facility statistics | Population-based, induced abortion | NA | ≥13 | 11.7 | NA |

(Continued)

**Table 5.** (Continued)

| Author, year | Country | Year of data | Type of data | Type of study sample/ population | Sample size (interviews conducted or denominator of proportion) | Weeks of gestation cutoff or range reported (at least ≥12 weeks) | Proportion of abortion later in pregnancy | Context and explanation for reported proportion of abortion later in pregnancy |
|---|---|---|---|---|---|---|---|---|
| Popinchalk & Sedgh 2019 [4] | Multi country, 22 countries reported | 2017 or most recent available | Administrative & facility statistics | Population-based, induced abortion | NA | Variable between countries, but reported as ≥13 in the paper | Range: 3.0–18 (Germany & The Netherlands respectively) | The authors explain that in Canada, the available data suggest that a large proportion of abortions are obtained after 13 weeks, but data are only available for abortions performed in hospitals, where abortions after 13 weeks are likely overrepresented. Additionally, a country's gestational age limits for legal abortion can affect the distribution of gestational age of abortion within a country, and in neighboring countries, if women have to travel to seek an abortion after 13 weeks. For example, there is no gestational age limit for legal abortion in the Netherlands, and the large proportion of abortions done after 13 weeks in the Netherlands are partly due to abortions obtained by non-residents. Finally, the prevalence of conscientious objection likely delays women's access to services. In Italy, a country with a lower proportion of abortions at less than 9 weeks gestation, the prevalence of conscientious objection has led to a shortage of abortion providers. |
| Restrepo-Bernal et al., 2019 [75] | Colombia | 2013–2014 | Medical records, 1 facility | Record review of all clients presenting for induced abortion | 87 | 18–26 | 19.6 | NA |
| Van de Velde et al., 2019 [76] | Belgium | 2013–2016 | Medical records, hospital system | Record review of all clients presenting for induced abortion | 28,741 | Two categories: 1. between 13 weeks + 2 days and 13 weeks + 6 days. 2. ≥14 2 | 1.0 & 2.4 respectively | NA |
| Fuentes et al., 2020 [77] | USA | 2012 & 2014 | Facility administered survey of clients, 8 facilities | Facility-based convenience sample, induced abortion | 721 | 13–15 & ≥16 | 6.3 & 3.9 respectively | NA |
| Goyal et al., 2020 [78] | USA | 2015–2016 | Medical records, 8 facilities | Record review of all clients presenting for induced abortion | 24,555 | 12–14; 15–17; & 18–24 weeks | 10.7, 5.3, & 3.2 respectively | NA |

(Continued)

**Table 5.** (Continued)

| Author, year | Country | Year of data | Type of data | Type of study sample/ population | Sample size (interviews conducted or denominator of proportion) | Weeks of gestation cutoff or range reported (at least ≥12 weeks) | Proportion of abortion later in pregnancy | Context and explanation for reported proportion of abortion later in pregnancy |
|---|---|---|---|---|---|---|---|---|
| Kebede et al., 2020 [79] | Ethiopia | 2019–2020 | Facility administered survey of clients, 1 facility | Facility-based convenience sample, induced abortion | 238 | 13–20 & >20 | 15.1 & 38.2 respectively | This study took place at one of the largest tertiary referral hospitals in Ethiopia. Additionally, the authors note the clinic associated with the hospital is the most well-organized family planning and abortion clinic in the country where service is provided by resident physicians and expert subspecialists. Most private institutions, health centers, and non- governmental organizations are not providing abortion care beyond 12 weeks |
| Shapiro et al., 2020 [80] | USA | 2012–2016 | Administrative statistics, vital statistics | Population-based, induced abortion | 137,128 | ≥20 | 2.0 | NA |
| Sharma & Pradhan 2020 [81] | India | 2015–2016 | Population-based survey | Population-based, induced abortion | 6,876 | ≥20 | 3.1 | NA |
| Moseson et al., 2021 [82] | USA | 2019 | Online survey | Purposive sample, induced abortion, trans and gender expansive people | 67 | 13–15, 16–20, & 21–24 | 6.0, 0, & 2.0 respectively | This study is from a small sample of trans and gender expansive people and not representative of other populations. |
| Mouba et al., 2021 [83] | Gabon | 2014–2018 | Medical records, 1 facility | Facility-based sample, induced abortion, clandestine with complication | 128 | ≥12 | 53.3 | This study included a small sample of clandestine abortion, majority misoprostol alone medication abortions. It's possible that misoprostol alone abortions in later stages are more likely to result in an abortion in process/incomplete, prompting people to come to a facility where they are counted (as opposed to <12 week where pregnancy is passed at home). |
| Schummers et al., 2022 [84] | Canada | 2017–2020 | Administrative statistics | Population-based, induced abortion | 84,032 | Second trimester, ≥14 | 5.1 | NA |
| Trapani et al., 2022 [85] | Brazil | 2014–202 | Medical records, 1 facility | Facility-based convenience sample, induced abortion after sexual violence | 141 | 12–20, >20 | 14.2 & 0.7 respectively | This analysis reports that 16.8% of people who sought abortion, the abortion did not occur either due to noncompliance with the protocol or due to stage of gestation. Later gestations were associated with the abortion not occurring. |

(Continued)

**Table 5.** (Continued)

| Author, year | Country | Year of data | Type of data | Type of study sample/population | Sample size (interviews conducted or denominator of proportion) | Weeks of gestation cutoff or range reported (at least ≥12 weeks) | Proportion of abortion later in pregnancy | Context and explanation for reported proportion of abortion later in pregnancy |
|---|---|---|---|---|---|---|---|---|
| White et al., 2022 [86] | USA | 2018 | Facility statistics, 12 facilities | Facility-based convenience sample, induced abortion | 4,455 | 12–15, ≥16 | 17.1 | This analysis was of Mississippi residents who obtained an abortion in 2018, 17% occurred at ≥12 weeks' gestation. This proportion may be contributed, in part, to Mississippi's restrictive policy and limited service environment. Additionally, 40% of all Mississippi patients traveled out of state for care, a much larger share than observed nationally. This reflects limited geographic accessibility of in-state options for many residents, and particularly for those needing care later in pregnancy. |
| Gonzolez-Perez et al., 2023 [87] | Colombia | 2015–2021 | Medical records, 5 facilities | Facility-based convenience sample, induced abortion | 20,423 | ≥15 | 12.1 | NA |
| Jubert et al., 2023 [88] | The Netherlands | 2020 | Facility administered survey of clients, 1 facility | Purposive sample, induced abortion, needed to travel | 35 | traveling because of gestational limit, >16 reported | 71.4 | This sample was of people who needed to travel for care based on gestational limits in France. |
| Malik et al., 2023 [90] | India | 2019–2021 | Population-based survey | Population-based, induced abortion | 5856 | Second or third trimester, unspecified | 33 | Women are often reluctant to report abortions due to stigma and discrimination, the number of abortions and possible reasons could be underreported in the dataset. Abortions later in pregnancy are more visible to the health system. Additionally, prior work has demonstrated that only 22% of abortion annually in 2015 occurred within facilities, [93] making this proportion of abortion later in pregnancy not representative of the population. |

care extend those delays further. Reproductive awareness such as knowledge of menstrual cycle, knowing how one gets pregnant, what to expect with using contraceptive methods, and when/how to test is critical to pregnancy detection [95]. Adolescents and young people are less likely to recognize or experience early symptoms of pregnancy [96], resulting in delays in care seeking. Delays in care can also lead to denial of services due to legal restrictions on gestational limits. These restrictions can compound and extend delays further when people need to travel to access care [2, 48, 70, 76, 80]. Previous work has shown that denial of abortion can have negative consequences on health, social, and economic outcomes [97–99].

Our findings of a wide range of estimates of the proportion of abortions that are later in pregnancy highlight a key gap: we lack standard terminology, definitions, and measures for abortion later in pregnancy both in population- and facility-based analyses. In addition, population-based data are sparse; abortion incidence data are challenging to collect, especially in legally restricted settings and for abortion later in pregnancy [100]. Caution must be used in interpreting and comparing results from different studies. Quantitative reports of abortion later in pregnancy do not share common definitions or standardized categories when reporting duration of gestation, and some studies use trimesters or months gestation as opposed to weeks. Trimester language is not preferred because it lacks specificity and is clinically not useful [101]. Differential definitions, terminology, and measures make it impossible to directly compare studies and difficult to interpret findings across studies and settings.

Facility-based studies of the proportion of abortions later in pregnancy present distinct limitations. The studies we included that presented facility-based proportions of abortions later in pregnancy were based on nonrepresentative samples of people who were admitted to facilities or who sought post-abortion care for a suspected complication. Facility-based samples were biased both towards higher proportions of abortion ≥12 weeks and higher incidence of complications, because, by definition, only abortions that resulted in facility-based care for complications or concerns were being counted. Abortions that occurred at later weeks of gestation that were uncomplicated were not recorded in these studies. Although the risk of complications increases with weeks of gestation [102], abortions later in pregnancy are safe [103], and abortion at any stage in pregnancy is safer than childbirth [104]. In addition, often a limited number of facilities have the capabilities to provide abortion care at later gestations, so facility data were often over representative of people who had traveled and those with more significant care needs. For example, a study we included from Ethiopia reported high proportions of abortions at 13–20 and >20 weeks of gestation (15.1% & 38.2% respectively), but also noted that the facility was a large tertiary hospital that received many referrals and complicated cases [79]. Therefore, caution must be taken when interpreting facility-based reports of proportions of abortion at later weeks of gestation; they are not the same as population-level estimates.

Legal restrictions on abortion significantly impede the collection of comprehensive high-quality data at the facility and population levels [100]. Individuals are unlikely to disclose their experiences due to stigma and fear of legal repercussions and indirect methods used to measure abortion incidence do not account for pregnancy duration [105]. These barriers contribute to the scarcity of population-based data and standardized measures for abortion later in pregnancy, as highlighted by the variability and lack of consistency in existing studies in this review. Addressing these data challenges and advocating for universal access to sexual and reproductive health services is important for understanding evidence gaps, enhancing the quality of care, and ensuring informed policy making that supports reproductive health and rights globally.

This scoping review must be interpreted with some limitations in mind. This review relied on published reports of proportions of abortions later in pregnancy; we did not access large scale population-based surveys or official data sources of ministries of health, therefore no

conclusions can be drawn about the prevalence of abortion ≥12 weeks and we do not calculate incidence of abortions. However, we do highlight gaps in data variability and availability in the literature and suggest the need for a unified approach for categorizing abortion later in pregnancy in future research. Data extraction was conducted by one author (LJ), although the team met regularly and discussed the inclusion/exclusion of papers when there was uncertainty. We excluded studies that were solely focused on spontaneous abortion, those that did not distinguish between induced and spontaneous abortion, and studies that focused only on abortion for non-viable pregnancies or congenital anomalies; however, this may result in under reporting because some people who present for induced abortion care may not report it as such due to stigma and fear of legal repercussions. This scoping review also has strengths: a standardized, transparent, and comprehensive multi-database search, and the inclusion of quantitative and qualitative studies published in English, Spanish, French, or Portuguese.

## Conclusion

The majority of abortions occur early in pregnancy. Circumstances surrounding abortions occurring later in pregnancy include health system challenges, late pregnancy recognition, financial issues, and delayed decision-making which are often intersecting and compounded, resulting in extended delays as weeks of gestation increase. Young people and those with lower levels of income were over-represented among those who need abortion later in pregnancy. Policies should facilitate access to safe abortion care with minimal delays. More research guided by clear methods and standard definitions when reporting on population and facility-based proportions of abortions later in pregnancy is needed to better inform policies and programs.

## Supporting information

**S1 Table. Bibliographic database search strategy for OVID/PubMed.**
(XLSX)

**S2 Table. Grey literature sources.**
(DOCX)

## Acknowledgments

The authors wish to acknowledge Oregon Health & Science University librarian Laura Zeigen for assistance conducting the literature searches.

## Author Contributions

**Conceptualization:** Bela Ganatra.

**Data curation:** Laura E. Jacobson.

**Formal analysis:** Laura E. Jacobson.

**Funding acquisition:** Bela Ganatra.

**Supervision:** Blair G. Darney, Bela Ganatra.

**Writing – original draft:** Laura E. Jacobson.

**Writing – review & editing:** Blair G. Darney, Heidi Bart Johnston, Bela Ganatra.

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
