## [Editor Report · Decision Letter 0]

10 Jul 2024

PGPH-D-24-01124

A global scoping review of the circumstances of care seeking for abortion later in pregnancy

Dear Dr. Jacobson,

Thank you for submitting your manuscript to PLOS Global Public Health. After careful consideration, we feel that it has merit but does not fully meet PLOS Global Public Health’s publication criteria as it currently stands. Therefore, we invite you to submit a revised version of the manuscript that addresses the points raised during the review process.

We look forward to receiving your revised manuscript.

Kind regards,

Camila Gianella Malca, Phd

Academic Editor

Journal Requirements:

Additional Editor Comments (if provided):

This paper addresses a critical issue that deserves more attention from researchers. The authors' efforts and the relevance of publishing research on this issue in academic journals are significant and their work is making a valuable contribution to the field.

While recognising the paper's relevance, it is also important to stress the challenges of performing a scoping review, such as the one presented in the article, of an issue highly dependent on the legal framework. The grounds for legal abortion are different in different contexts. Therefore, the internal validity of the analysis must explain how the authors take into account the legal framework of each of the countries included in the review. This is critical because what is called an administrative or bureaucratic barrier in one country could be related in another with a more restrictive ground for legal abortion, the result of the fair to criminalisation.

It is also essential to analyse, or at least consider, the legal reforms performed in the countries to guarantee or block access to safe abortion. For example, in India, while it is true that abortion has been legal since 1971, only in 2003 did the country implement a legal reform (issued a regulation) to improve access to free abortion services. In Mozambique, abortion has been legal since 2014, and in Colombia, the legal grounds for abortion have dramatically changed in recent years due to the decisions of the Constitutional Court.

Considering this, I ask the authors to review the article and explain how they have addressed the critical issue of the legal status and legal grounds for abortion in each of the studies included. For example, the legal grounds for abortion in the country where the study selected was performed were considered an inclusion/exclusion variable (the authors state that the research doesn’t contain a control variable), and how the legal grounds could affect the results.

It will also be helpful to explain the rationality behind the time frame, considering that the selected articles included registers from the 90s.

Best regards,
---

## [Editor Report · Decision Letter 1]

13 Aug 2024

PGPH-D-24-01124R1

A global scoping review of the circumstances of care seeking for abortion later in pregnancy

Dear Dr. Jacobson,

Thank you for submitting your manuscript to PLOS Global Public Health. After careful consideration, we feel that it has merit but does not fully meet PLOS Global Public Health’s publication criteria as it currently stands. Therefore, we invite you to submit a revised version of the manuscript that addresses the points raised during the review process.

We look forward to receiving your revised manuscript.

Kind regards,

Camila Gianella Malca, Phd

Academic Editor

Journal Requirements:

Additional Editor Comments (if provided):

This paper addresses a critical issue that deserves more attention from researchers. The authors' efforts and the relevance of publishing research on this issue in academic journals are significant and their work is making a valuable contribution to the field.

While recognising the paper's relevance, it is also important to stress the challenges of performing a scoping review, such as the one presented in the article, of an issue highly dependent on the legal framework. The grounds for legal abortion are different in different contexts. Therefore, the internal validity of the analysis must explain how the authors take into account the legal framework of each of the countries included in the review. This is critical because what is called an administrative or bureaucratic barrier in one country could be related in another with a more restrictive ground for legal abortion, the result of the fair to criminalisation.

It is also essential to analyse, or at least consider, the legal reforms performed in the countries to guarantee or block access to safe abortion. For example, in India, while it is true that abortion has been legal since 1971, only in 2003 did the country implement a legal reform (issued a regulation) to improve access to free abortion services. In Mozambique, abortion has been legal since 2014, and in Colombia, the legal grounds for abortion have dramatically changed in recent years due to the decisions of the Constitutional Court.

Considering this, I ask the authors to review the article and explain how they have addressed the critical issue of the legal status and legal grounds for abortion in each of the studies included. For example, the legal grounds for abortion in the country where the study selected was performed were considered an inclusion/exclusion variable (the authors state that the research doesn’t contain a control variable), and how the legal grounds could affect the results.

It will also be helpful to explain the rationality behind the time frame, considering that the selected articles included registers from the 90s.
---

## [Decision Letter · Decision Letter 2]

31 Oct 2024

A global scoping review of the circumstances of care seeking for abortion later in pregnancy

PGPH-D-24-01124R2

Dear MS Jacobson,

We are pleased to inform you that your manuscript 'A global scoping review of the circumstances of care seeking for abortion later in pregnancy' has been provisionally accepted for publication in PLOS Global Public Health.

Best regards,

Camila Gianella Malca, Phd

Academic Editor

The article is ready for publication

Reviewer Comments (if any, and for reference):

Reviewer's Responses to Questions

**Comments to the Author**

1. If the authors have adequately addressed your comments raised in a previous round of review and you feel that this manuscript is now acceptable for publication, you may indicate that here to bypass the “Comments to the Author” section, enter your conflict of interest statement in the “Confidential to Editor” section, and submit your "Accept" recommendation.

Reviewer #1: All comments have been addressed

Reviewer #2: All comments have been addressed

2. Does this manuscript meet PLOS Global Public Health’s publication criteria? Is the manuscript technically sound, and do the data support the conclusions? The manuscript must describe methodologically and ethically rigorous research with conclusions that are appropriately drawn based on the data presented.

Reviewer #1: (No Response)

Reviewer #2: Yes

3. Has the statistical analysis been performed appropriately and rigorously?

Reviewer #1: (No Response)

Reviewer #2: N/A

4. Have the authors made all data underlying the findings in their manuscript fully available (please refer to the Data Availability Statement at the start of the manuscript PDF file)?

Reviewer #1: (No Response)

Reviewer #2: Yes

5. Is the manuscript presented in an intelligible fashion and written in standard English?

Reviewer #1: (No Response)

Reviewer #2: Yes

6. Review Comments to the Author

Reviewer #1: (No Response)

Reviewer #2: An interesting, well -written article addressing a critical issue worth publication. I believe previous reviewer comments have been appropriately addressed with the addition of specific information on the legal status of abortion in the countries of the included studies and the paragraph added to the text. While it would be fascinating to see an analysis that compared how factors vary between countries in which abortion is legal and those in which it is illegal, there is not enough of a black/white distinction between those countries to allow that kind of analysis. Just because abortion is legal in a country does not necessarily mean it will be easy to access these services and vice versa. There are too many other factors in play. Maybe something that can be tackled in future research….

My one concern was the exclusion of late term abortions due to what the authors euphemistically call “receiving new medical information,” thus non-viability of the fetus or congenital anomalies. They claim that they excluded these studies because abortion for these causes “is well documented in the literature, more clearly understood, and less stigmatized than other circumstances.” While this might be true, the United States is a vivid example of how even abortion justified by medical reasons can be subject to, in some cases, fatal delays (the now famous “sending of women to the parking lot till they start bleeding out,” so their care can be considered emergency care, not an abortion). I agree that it would be too much work to include data on abortions for medical reasons at this point, but maybe the authors could at least provide a rough % of what later-term abortions fall into that category.

7. PLOS authors have the option to publish the peer review history of their article (what does this mean?). If published, this will include your full peer review and any attached files.

**Do you want your identity to be public for this peer review?** For information about this choice, including consent withdrawal, please see our Privacy Policy.

Reviewer #1: No

Reviewer #2: No
